# Reciprocal Effects of Metal Mixtures on Phytoplankton

**Ammara Nawaz, Pavlína Eliška Šotek and Marianna Molnárová ***

Department of Environmental Ecology and Landscape Management, Faculty of Natural Sciences, Comenius University Bratislava, Mlynská dolina, Ilkovičova 6, SK-842 15 Bratislava, Slovakia; nawaz3@uniba.sk (A.N.); liscakova5@uniba.sk (P.E.Š.)
* Correspondence: marianna.molnarova@uniba.sk

**Abstract:** Several types of contaminants are anthropogenically introduced into natural aquatic ecosystems and interact with other chemicals and/or with living organisms. Although metal toxicity alone has been relatively well studied, the toxic metal ion effects in the mixture have been thoroughly studied only during the last decades. This review focuses on the published reciprocal effects of different metals on different species of algae, together with describing their toxic effects on studied parameters. Phytoplankton as a bioindicator can help to estimate the reciprocal metal risk factor. Many methodologies have been developed and explored, such as the biotic ligand model (BLM), concentration addition (CA), independent action (IA), sensitivity distribution of EC50 species sensitivity distribution (SSD curves), and others, to study reciprocal metal toxicity and provide promising results, which are briefly mentioned too. From our review, we can commonly conclude the following: Zn acted antagonistically with most heavy metals (Al, Cu, Cd, and Ni). The Cu interaction with Cd, Fe, and Pb was mostly antagonistic. Cd showed synergistic behaviour with Hg, Cu, Zn, and Pb and antagonistic behaviour with Co and Fe in many cases. Methods and techniques need to be developed and optimised to determine reciprocal metal toxicity so that the ecotoxicological predictions made by using phytoplankton can be more accurate and related to real-time toxic metals risks to the aquatic ecosystem. This is the main objective of ecotoxicological tests for risk assessment. Understanding how metals enter algal cells and organelles can help to solve this challenge and was one of the main parts of the review.

**Keywords:** phytoplankton; reciprocal toxicity; toxic metals; aquatic ecosystem



## 1. Introduction

Water quality is one of the major factors that affects the health status of all the life forms present in aquatic system. Water contamination is indeed a serious hazard to the world and mankind. Anthropogenic activities are one of the major causes of water pollution, which expel multiple kinds of harmful pollutants into aquatic ecosystems. Different pollutants (such as toxic metals, dyes, pharmaceuticals, chemical waste, organic compounds, and nanoparticles) have diverse effects on living cells and the environment depending on their type and origin. The contamination of the aquatic environment with heavy metals is a critical issue nowadays due to the potential toxicological risks and accumulative nature of metals in aquatic systems [1]. Contributing sources of heavy/toxic metal contamination include, e.g., soil erosion, weathering of the Earth's crust, metal industries, chemical industries, mine waste, the battery manufacturing industry, leather tanning effluents, fertilizer industries, and paint industries. The properties of the given metal and the given environmental factors naturally influence the distribution of metals in the environment. The mobilisation of heavy metals has increased considerably in natural systems due to human interference. Some metals are essential for the biochemical and physiological processes that occur in living cells in very low or controlled concentrations; however, higher concentrations of these metals above a threshold limit result in cell toxicity and

damage on many levels. The indirect effect of heavy metal pollution on the structure and functions of food webs in different ecosystems, including aquatic ones, is observable [2].

Phytoplankton are a key part of marine and freshwater ecosystems as primary producers of the food web. They are exceptional in their role to convert carbon dioxide as a source of inorganic C into oxygen and organic C via photosynthesis. Phytoplankton are mostly microscopic, single-celled photosynthetic organisms that live near the surface of water, where sunlight is enough for them. They are very diverse, varying from photosynthesising bacteria to plant-like algae or armour-plated coccolithophores, and, in addition, they involve important groups such as diatoms, cyanobacteria, and dinoflagellates. Some of these organisms' forms occasionally bloom in response to changing seasons in relation to the bioavailability of nutrients (e.g., N, Fe, and P). Due to their faster turnover rates than higher plants, phytoplankton can respond rapidly not only to climate variations, but also to the contamination of water.

In this review, the toxicity and damage induced by metals in mixtures (reciprocal effects) to phytoplankton will be discussed briefly, with reference to previous investigations. In natural ecosystems, waste products along with many heavy metals in different concentrations are discharged in mixtures, posing a real threat to aquatic life, humans, and the environment. It is important today to study this aspect of metal toxicology in detail and make logical predictions for the safety not only of aquatic ecosystems, but of the world.

## 2. Phytoplankton as Heavy Metal Pollution Indicators

The phytoplanktonic community is considered to be a basic autotrophic part of any aquatic ecosystem, affecting the assembly and efficiency of the food web of the system. Phytoplankton also affect global biogeochemical cycles, as documented in a recent study on the role of oceanic phytoplankton in the C, P, and N cycles and their distribution [3,4]. These authors demonstrated the importance of particulate N:C and P:C ratios for the regulation of dissolved inorganic matter (dN:P) on the global scale, with the level of marine oxygen being an important control [4]. Their research provides additional information on the potential interdependence of phytoplankton physiology and global climate conditions. Hence, phytoplankton are used as an early warning signal for the health status of water bodies [5]. The variability of metals in the phytoplanktonic community could be used to predict the intensity and potential of heavy metal ecological damage and water quality decline on many levels [6]. Algal parameters such as growth, chlorophyll content, photosynthesis, metal uptake and metabolism, or lipid profile are commonly determined to find out the level of stress in algal cells resulting from water pollutants, including heavy metals [7–10]. Lewis [11] mentioned that phytoplankton are good ecotoxicological tools, as they are the main autotrophic component of the water system. Most phytoplankton species exhibit a short life span with a response of very high sensitivity to different environmental fluctuations, including pollutants such as heavy metals [11]. Furthermore, phytoplankton are easily and economically culturable and show rapid growth and cellular turnover with high sensitivity levels for various pollutants. Phytoplankton are being used as bioindicators for heavy metal toxicity studies in aquatic ecosystems, providing valuable results and insights [12].

Phytoplankton can accumulate a certain amount of metals without being damaged, and this process is termed phytoremediation; however, exposure to a high amount of heavy metals usually results in damaging the living cells [13]. Algal species vary in their individual metal sensitivities and interactions of the metal mixture interactions [14], and some studies are focused on studying the resistant strains of freshwater algae [15].

Upon heavy metal exposure, phytoplankton homeostasis disruption can occur. In stress conditions, algal cells are observed to produce excessive reactive oxygen species (ROS), for example, superoxide ($O_2 \cdot^-$), hydroxyl radical ($\cdot OH$), or hydrogen peroxide ($H_2O_2$). ROS can be damaging to the proteins, amino acids, nucleic acids, membrane lipids, and DNA of phytoplankton, causing several disorders in the algal cell [16,17]. The antioxidant protection enzymatic and nonenzymatic system, like superoxide dismutase

(SOD), peroxidase (POD), catalase (CAT), glutathione reductase (GR), and very effective sulfur-rich molecules of phytochelatins/metallothioneins, is released by the photosynthetic cell to decrease the excess ROS induced by metal exposure, to overcome the damage against the given pollutant. Metallothioneins and phytochelatins produced in the cytosol are abundant intracellularly and extracellularly to bind metal ions to the exudate, precipitate, and stabilize them on the cell surface to prevent their cellular entry into microalgae [18]. To decrease cellular oxidative damage induced against the applied metal, the accumulation of free proline in the cell is also noticed in heavy-metal-stressed algal cells [19]. Indeed, higher concentrations of, for example, Cu and Cr, were inhibitory to proline accumulation by *Chlorella vulgaris* [20]. It is well documented that, in metal-stressed-state algal and phytoplanktonic cells, malondialdehyde (MDA) and TBA-reactive products (TBARS, thiobarbituric acid reactive products) are produced that are frequently used as metal stress indicators in ecotoxicological evaluation studies.

Zheng et al. [21,22] studied the geochemical behaviour of heavy metals in the water system. They explained that heavy metals can show three arrangements when released into a water body as,

a. Particle form: The metal is adsorbed onto the suspended particles already present in the water body.

b. Dissolved form: Metals bond with dissolved organic materials available in the water system.

c. Biological form: The metal is taken up by the phytoplankton and algae. It is integrated in the cell and passed on to the food chain.

The dynamics of heavy metals in a water body are primarily controlled by these three metal arrangements. Heavy metal bioaccumulation in the food web of aquatic systems is mainly controlled by the amount of metal taken up by the phytoplankton community. This makes phytoplankton a good bioindicator for determining the metal toxicity or health of any aquatic ecosystem and they are used extensively in ecotoxicological studies.

## 3. Behaviour of Heavy Metals in an Aquatic Ecosystem

Heavy metals are often found in nature as a mixture with other pollutants. Waste materials such as industrial effluents, mining waste, agricultural waste, and sewage waste drained through multiple sources into the same area contain both organic and inorganic compounds that interact with each other, altering the toxicity and nature of contaminants, including heavy metals. Therefore, it is essential to evaluate the toxicity and risk posed by these elements in mixture form to obtain a more reality-based evaluation of metal toxicity [23]. Understanding the synergistic effects for metal co-exposure in ecotoxicology is required to evaluate the real-time toxicity of metals, as, in natural ecosystems, pollutants are discharged in the form of a bulk. In nature, organisms in any environment experience exposure to mixtures of metals, while the ecotoxicity of metals alone is different compared to a mixture of different metals [23,24]. Limited studies are available to evaluate metal toxicity as a reciprocal effect. Today, most toxicological research is focused on the study of metal stress in terms of synergistic, additive, or antagonistic effects.

Different environmental conditions affect the reciprocal toxicity of heavy metals, together with the different properties of toxic elements, as well as their interactions with organic and inorganic substances outside and inside of living cells [23]. These interactions include metal speciation, binding with other ligands, and the mechanisms of transport of metals. Absorption, dispersal, metabolism, and detoxification are other phenomena that happen differently in different species, making it difficult to explain the toxicity of any metal in general. Therefore, deeper investigations are mandatory. These interactions can be additive (sum of individual toxicity), interactive (synergistic or antagonistic), or independent of each other, which can produce different unpredicted responses in the test species.

The bioaccumulation of heavy metals in plankton cells depends on multiple factors, including the absorptive ability of individual strains, season, pH, temperature, metal

bioavailability, and many others [25,26]. The uptake of As by microalgae is strongly influenced by $(PO_4)^{3-}$ in the culture medium, as confirmed in *Anabaena* sp. [27], as arsenate reduction was observed with high phosphate concentrations in batch culture experiments. This is because As, especially $(AsO_4)^{3-}$, competes with inorganic P transporters. Atici et al. [28] observed the seasonal variation in the patterns of heavy metal accumulation and the stress of the phytoplankton in a freshwater body, Sariyar Dam Lake in Turkey. The heavy metal accumulation pattern in plankton samples in all four seasons was the same as Pb > Cr > Cd > Hg, in agreement with the bioavailability of these metals in the same order in the lake water samples of the freshwater system, showing a directly proportional relationship between metal abundance and bioabsorption by phytoplankton. However, the amount of metal taken up by the phytoplankton cells varied in each season (different pH, temperatures, and metal availability). The highest metal uptake was observed in the winter (Cd and Cr) and spring (Pb) seasons. Therefore, metals in combination in a water body accumulate in phytoplankton cells differently under different conditions.

## 4. Conventional and Novel Methods to Study Reciprocal Metal Mixture Toxicity Used Frequently

Now, scientists commonly apply a statistical approach of relative metals or ionic toxicities to predict the relative toxicity of metals and their possible reciprocal interactions in combined form. It is known that metal toxicity for algae can be eliminated by sorption or adsorption to the cell wall of the coenobia. Therefore, many authors determine the metal effect on uptake alone or in mixtures of other metals such as extracellular (membrane-bound) and intracellular metal concentrations [29]. There is also software that is used to simulate the chemical composition in solutions with contact with solid compounds, particle surfaces, or other inorganic ions, e.g., Visual MINTEQ v.4 (https://vminteq.com/, accessed on 5 February 2024) or Dynamic Energy Budget (DEB) software, focusing on solving (eco)toxicological problems (DEBtox) replaced by Bring/Build Your Own Model (BYOM) with its flexible set of Matlab scripts and functions (https://www.debtox.info, accessed on 23 June 2022). The toxicokinetic–toxicodynamic (TK-TD) model is used because it simulates the toxicity process at the level of the organism over time and can quantify and predict the toxicity of metals and organic pollutants [30,31].

There are more articles, including reviews, that are focused on performing a risk assessment and finding the best-fitting modelling calculations of contaminants in aquatic environments, as well as the toxicity of contaminants to aquatic species, including algae, together to determine the relevant important processes and conditions/parameters related to chemical toxicity to organisms [32–36]. Gregorio et al. [37], in their study, compared two methodologies used to determine the combined toxicity of pollutants (chemicals/heavy metals, etc.). One of the first methods is called **concentration addition** (the CA model is applied to chemicals with similar modes of action) or the response addition method, which is used considering the species sensitivity distribution (SSD) curves. The ecotoxicological estimate made by this method seems to be reliable only when it is applied to a single species model. In the case of multiple species simultaneously aggregated to SSDs, the results generated by this method are not very consistent. De Zwart and Posthuma [38] supported the use of the concentration and response addition method to determine the reciprocal toxicity in single or multiple species. Backhaus et al. [39] seriously questioned the application of **independent action** (the IA model is applied to chemicals with dissimilar modes of action) and the **concentration addition** (CA) method in ecotoxicological studies of freshwater and marine ecosystems for combined risk assessments. Sometimes, a **bio-concentration factor** (BCF) and a **bioaccumulation factor** (BAF) are another possibility for assessing metal toxicity [40]. However, a high variability in the mean BCF/BAF values of all metals has been observed, and this very high variability remains in the chronic exposure range of the metals, indicating that real-time metal toxicity is very dependent on the metal uptake ability of the biota, the detoxification ability of individual species, and many other factors, making such assessments not very accurate. Another option to determine the

risk assessment of toxic pollutants in mixtures is to use available data based on the **EC50** values that are mostly available for the indicator organism and apply **the concentration addition model** to obtain a toxicological estimation [41]. If this initial assessment shows the acute toxicity of a given pollutant (metal, chemical, or any other), a detailed study can be planned to determine the exact nature of, toxicity, and possible damage to this pollutant. The predicted environmental concentration/predicted no environmental concentration ratios (PEC/PNEC) can also be helpful in such initial tests. The PNEC is calculated for each component and the PEC/PNEC ratio of all individual components is summed to determine the final risk quotient (RQ) of the pollutant mixture, including heavy metals [42].

Phytoplankton community responses can also be used to evaluate the ecological impacts of heavy metals. In research conducted by Kapkov et al. [43], the response of the phytoplankton community was observed under Cu, Co, and Cd stress conditions, isolated from the White Sea. Different algal strains behaved differently with reciprocal and individual exposure to metals. Changes in growth, pigments, and morphology were observed. *Chaetoceros radicans* and *Ditylum brightwellii* showed the highest sensitivity to heavy metals. Many anomalies in morphology and size could be detected in both species. *Thalassionema nitzschioides* and *Chaetoceros curvisetus* were reported to be highly resistant to these metals. A significant change in the number of dominant species and composition was observed in the structure of the phytoplankton community after exposure. As stated in Table 1, cell growth inhibition was different in different species. This can explain why different phytoplankton behave differently towards applied metal stress and the community structure behaves in a different way compared to a single algae cell exposed to certain metals or metal mixtures. The photosynthetic activity of diatoms was higher, even under low light conditions, compared to green algae; for example, the marine diatom *Pheodactyulum tricornutum* showed a higher photosynthetic efficiency than *Chlorella vulgaris* in low light. Diatoms, both photoprotective and light-harvesting pigments, are synthesised from the same precursors, and the α-carotene biosynthetic pathway is absent in them in contrast to chlorophytes [44]. Upon metal exposure, diatoms show community cellular effects which can be used as indicators for biomonitoring [45]. Growth inhibition, pigment depletion, and oxidative stress are seen in algal groups and diatoms upon metal application. Phytochelatin in *Thalassiosira weissflogii* was observed to lower Cd toxicity in a high Cd environment [46]. Metal-binding polysaccharides have been studied in diatoms and algae to minimise stress effects [47].

**Table 1.** Effect of reciprocal metal exposure (Cd, Co, and Cu) on various algal populations [43].

| Metal Mixture | Algal Species with Highest Growth Rate | Algal Species with Lowest Growth Rate |
|---|---|---|
| Cd + Cu | *Chaetoceros curvisetus*, Flagellates ($7 \times 10^6$ cells L$^{-1}$) | *Navicula* sp., *Chaetoceros diadema*, *Chaetoceros radicans*, *Ceratium fusus* ($1 \times 10^6$ cells L$^{-1}$) |
| Cd + Co | *Chaetoceros curvisetus*, *Thalassionema nitzschioides*. Flagellates, *Chaetoceros diadema*, *Gymnodinium* sp. ($12 \times 10^6$ cells L$^{-1}$) | *Melosira nummuloides* ($1 \times 10^6$ cells L$^{-1}$) |
| Cu + Co | *Thalassionema nitzschioides*, Flagellates ($8 \times 10^6$ cells L$^{-1}$) | *Navicula* sp., *Chaetoceros radicans*, *Ditylum brightwellii*, *Gymnodinium* sp. ($1 \times 10^6$ cells L$^{-1}$) |
| Cd + Co + Cu | *Ceutorhynchus curvisetus*, *Thalassionema nitzschioides* ($13 \times 10^6$ cells L$^{-1}$) | *Chaetoceros diadema*, *Ceratium fusus*, Flagellates ($3 \times 10^6$ cells L$^{-1}$) |

The **biotic ligand model** (BLM) is another useful model for predicting the effects of metals towards aquatic life that are primarily focused on the interactions of heavy metals with the biological surfaces of the aquatic life, mainly with the binding sites of the biological membranes of cells [48]. Using multiple methods, in comparison, can be a good approach for making more precise ecotoxicological predictions of aquatic metal pollution using phytoplankton [42].

## 5. Previous Investigations Performed to Determine Reciprocal Toxicity of Metal Mixtures in Phytoplankton

Scientists and environmentalists have been attempting to examine the mixture toxicity of metals in aquatic systems. The objective of such studies is always to determine the possible risk implied by any metal mixture drained into a water body. Phytoplankton are usually used in such assessment tests, as they are a common and reliable indicator organism for the evaluation of aquatic ecosystem pollution evaluation. Table 2 summarises the ecotoxicological reciprocal metal assessments made using phytoplankton test species.

**Table 2.** A summary of reciprocal toxicity assessments made in previous studies using phytoplankton (EE—equivalent effect concentration; NA—not available).

| Metal Mixture | Species | pH | Concentrations (mg L$^{-1}$ or M) | Reciprocal Effect | Reference |
|---|---|---|---|---|---|
| Al + Zn | *Raphidocelis subcapitata* | NA | >0.026 mg L$^{-1}$ Zn and 0.739 mg L$^{-1}$ Al; 22.24–37.06 µM Al, 0.08–0.46 µM Zn | Antagonistic | [49,50] |
| Cu + Cr + Ni | *Chlorella pyrenoidosa* 251 | 6.8 | 0.1–1.0 mg L$^{-1}$ of Cu, Cr and Ni | Synergistic | [51] |
| As + Se | *Desmodesmus quadricauda* | 7.2 | 29.05 mg L$^{-1}$ As and 3.65 mg L$^{-1}$ Se | Synergistic | [16] |
| Cd + Co | *Chlamydomonas reinhardtii* | 7 | $2 \times 10^{-8}$ M Cd and Co | Non-interactive | [52] |
| Cd + Fe + Mn + Cu | *Chlamydomonas reinhardtii* | 7 | $2 \times 10^{-8}$ M Cd$^{2+}$, $1 \times 10^{-17}$ M Fe$^{3+}$, $1 \times 10^{-6}$ M Mn$^{2+}$, $1 \times 10^{-13}$ M Cu$^{2+}$ | Non-interactive | [52] |
| Cd + Co | *Chlorella vulgaris* | 6.5 | 0.89 µM Cd and 9.50 µM Co | Antagonistic | [53] |
| Cd + Cr | Nile river algal community | NA | 0.05–1.00 mg L$^{-1}$ Cd and 0.25–3.00 mg L$^{-1}$ Cr | Synergistic | [54] |
| Cd + Cu | *Chaetoceros gracilis*; *Isochrysis* sp. | NA | 0, 0.56, 1.00, 1.80, 3.20, and 5.60 mg L$^{-1}$ Cd and 0, 0.010, 0.018, 0.032, 0.056, 0.100 mg L$^{-1}$ Cu | Synergistic | [55] |
| Cd + Cu | *Chlamydomonas reinhardtii* | 7.5 | 40, 60, and 80 nM Cd and Cu | Antagonistic | [56] |
| Cd + Cu | *Chlamydomonas reinhardtii* | 8 | $1 \times 10^{-6}$–$1 \times 10^{-5}$ M Cd, and $1 \times 10^{-6}$–$1 \times 10^{-5}$ M Cu | Synergistic | [57] |
| Cd + Cu | *Chlamydomonas reinhardtii* | 6 | $3.52 \times 10^{-6}$ Cu$^{2+}$ M and $3.52 \times 10^{-6}$ M Cd$^{2+}$ | Antagonistic | [31] |
| Cd + Cu | *Chlorella pyrenoidosa* | 8.6 | 13–25 µM Cu and 6 µM Cd | Synergistic | [58] |
| Cd + Cu | *Chlorella vulgaris* | | | Antagonistic | [59] |
| Cd + Cu | *Chlorella vulgaris* | NA | 1.5 µM Cu and 2.0 µM Cd | Synergistic | [60] |
| Cd + Cu | *Chlorella vulgaris* | 6.5 | 2.80 µM Cu and 0.89 µM Cd | Synergistic | [53] |

**Table 2.** *Cont.*

| Metal Mixture | Species | pH | Concentrations (mg L$^{-1}$ or M) | Reciprocal Effect | Reference |
|---|---|---|---|---|---|
| Cd + Cu | *Chlorella* sp. | | | Synergistic | [29] |
| Cd + Cu | *Chlorolobion braunii* | NA | 5 μM Cu and 1 μM Cd | Synergistic | [61] |
| Cd + Cu | *Dunaliella minuta* | 7.4 | 7.57 μM Cu and 0.34 μM Cd | Antagonistic | [62] |
| Cd + Cu | *Navicula pelliculosa* | 7 | 0.42–0.54 μM Cu and 0.50–0.59 μM Cd (EC50 values) | Antagonistic | [32] |
| Cd + Cu | Nile river algal community | NA | 0.05–1.00 mg L$^{-1}$ Cd and Cu | Synergistic | [54] |
| Cd + Cu | *Pseudokirchneriella subcapitata* | 8.1 (BLM) | 0.006–0.046 μM Cu and 0–0.500 μM Cd | Synergistic | [33] |
| Cd + Zn | *Chlorella vulgaris* | 6.8 | $2 \times 10^{-5}$ M Zn and $0$–$8 \times 10^{-5}$ M Cd | Antagonistic | [24] |
| Cd + Fe | *Thalassiosira weissflogii* | NA | $1 \times 10^{-10}$ M Cd$^{2+}$ and $1 \times 10^{-7.8}$ to $1 \times 10^{-5.8}$ M Fe EDTA | Antagonistic | [63] |
| Cd + Hg | *Anabaena inaequalis* | NA | | Synergistic | [64] |
| Cd + Ni | *Anabaena inaequalis* | NA | | Antagonistic and synergistic depending upon metal conc. | [64] |
| Cd + Pb | *Scenedesmus obliquus* | NA | EE-20 for Cd-Pb synergistic, EE-50 additive | Synergistic | [65] |
| Cd + Zn | *Chlamydomonas reinhardtii* | 7 | $1 \times 10^{-9}$ M Zn$^{2+}$, $7 \times 10^{-9}$ M Cd$^{2+}$ | Antagonistic | [66] |
| Cd + Zn | *Chlamydomonas reinhardtii* | 7 | 7 nM Cd$^{2+}$ and $6 \times 10^{-9}$ M | Antagonistic | [52] |
| Cd + Zn | *Chlorella* sp. | | | Antagonistic | [29] |
| Cd + Zn | *Skeletonema costatum* | 7.8 to 9 | 200–400 μg L$^{-1}$ Zn 100 μg L$^{-1}$ Cd | Additive to slight synergistic | [67] |
| Cd + Zn | *Phaeodactylum tricornutum* | 7.8 to 9 | 3000 μg L$^{-1}$ Cd 4000 μg L$^{-1}$ Zn | Additive to slight antagonistic | [67] |
| Cd + Zn | *Scenedesmus obliquus* | NA | EE-20 and EE-50 for Cd-Zn additive | Synergistic | [65] |
| Co + Cu | *Chlorella vulgaris* | 6.5 | 9.5 μM Co and 2.8 μM Cu | Synergistic | [53] |
| Cu + Fe | *Chlamydomonas reinhardtii* | 6–8 | $1 \times 10^{-19}$ M Fe$^{3+}$ and $1 \times 10^{-13}$ to $1 \times 10^{-10.5}$ | Antagonistic | [68] |
| Cu + Ni | *Pseudokirchneriella subcapitata* | 6.2–8.2 | 0.001–2.680 mg L$^{-1}$ Ni and 0.001–0.659 mg L$^{-1}$ Cu | Non-interactive | [69] |
| Cd + Ca | *Micrasterias denticulata* | NA | 2 mM CaSO$_4$ and 150 μM CdSO$_4$ | Antagonistic | [70] |
| Cu + Pb | *Chlamydomonas reinhardtii* | 7 | ≤1 mg L$^{-1}$ of Cu and Pb | Antagonistic | [71] |

**Table 2.** *Cont.*

| Metal Mixture | Species | pH | Concentrations (mg L$^{-1}$ or M) | Reciprocal Effect | Reference |
|---|---|---|---|---|---|
| Cu + Zn | *Chlorella* sp. | | | Antagonistic | [29] |
| Cu + Zn | *Navicula pelliculosa* | 7 | 3.48 µM Zn and 0.51 µM Cu (EC50 values) | Additive | [32] |
| Cu + Zn | *Phaeodactylum tricornutum* | NA | 0.25 mg L$^{-1}$ Cu and 4.00 mg L$^{-1}$ Zn | Synergistic | [72] |
| Cu + Zn | *Phaeocystis antarctica*; *Cryothecomonas armigera* | 7.9 | | Antagonistic | [73] |
| Cu + Zn | *Scenedesmus* sp. | 7 | 2.5–40.0 µM CuCl$_2$.2H$_2$O and 5–100 µM ZnCl$_2$ | Synergistic | [19] |
| Cu + Zn | *Pseudokirchneriella subcapitata* | 8.1 (BLM) | 0.20–2.00 µM Zn and 0.006–0.046 µM Cu | Antagonistic | [33] |
| Cd + Zn | *Pseudokirchneriella subcapitata* | 8.1 (BLM) | 0.20–2.0 µM Zn 0.036–2.100 µM Cd | Antagonistic | [33] |
| Cr + Cu | *Chlorella vulgaris* | NA | 0.05, 0.50, 5.00 µM | Additive | [74] |
| Hg + Ni | *Anabaena inaequalis* | | | Additive | [64] |
| Mg + Pb | *Chlamydomonas reinhardtii* | 7 | ≤1 mg L$^{-1}$ of M and Pb | Antagonistic | [71] |
| Ni + Zn | *Navicula pelliculosa* | 7 | 0.15–0.19 µM Ni and 3.48–3.71 µM Zn (EC50 values) | Synergistic | [32] |
| P + Zn | *Raphidocelis subcapitata* | NA | $0.09 \times 10^{-6}$ M to $9.08 \times 10^{-6}$ M Zn and $2.3 \times 10^{-4}$ M, $2.3 \times 10^{-6}$ M and $1.0 \times 10^{-6}$ M P | Additive | [75] |
| Pb + Zn | *Scenedesmus obliquus* | NA | EE-20 and EE-50 for Pb-Zn synergistic | Additive | [65] |
| As(V) + Cd + Cu + Ni + Pb | *Diacronema lutheri* | NA | 450 µg L$^{-1}$ As(V), 109 µg L$^{-1}$ Cd, 34 µg L$^{-1}$ Cu, 126 µg L$^{-1}$ Ni, 414 µg L$^{-1}$ Pb | As(V) had the main toxicity in the mixture | [76] |
| Cd + Co | *Raphidocelis subcapitata* | NA | 0.13–0.25 mg L$^{-1}$ Co, 0.025–0.100 mg L$^{-1}$ Cd | Synergistic (high Co and low Cd) Antagonistic (low Co and high Cd) | [77] |
| Cd + Co + Cu | *Chlorella vulgaris* | 6.5 | 2.80 µM Cu, 0.89 µM Cd and 9.50 µM Co | Antagonistic | [63] |
| Cd + Cr + Cu | Nile river algal community | NA | 0.05 mg L$^{-1}$ Cd and 0.10 mg L$^{-1}$ Cu, Cr | Antagonistic | [78] |
| Cd + Ni + Zn | Nile river algal community | NA | 0.05 mg L$^{-1}$ Cd and 0.10 mg L$^{-1}$ Cu, Zn | Antagonistic | [78] |
| Co + Cu + Zn | *Chlorophyceare*; *Bacilariophyceae*; *Cyanophyceae* | NA | $1 \times 10^{-6}$ to $1 \times 10^{-10}$ mg L$^{-1}$ Cu, Co and Zn | Synergistic | [79] |
| Cu + Ni + Zn | *Pseudokirchneriella subcapitata* | 7.2 | 0.0200 mg L$^{-1}$ Zn, 0.0010 mg L$^{-1}$ Ni, 0.0025 mg L$^{-1}$ Cu | Non-interactive | [80] |

**Table 2.** *Cont.*

| Metal Mixture | Species | pH | Concentrations (mg L$^{-1}$ or M) | Reciprocal Effect | Reference |
|---|---|---|---|---|---|
| Cu + Ni + Zn | *Pseudokirchneriella subcapitata* | 6.2–8.2 | 0.001–2.680 mg L$^{-1}$ Ni, 0.001–0.659 mg L$^{-1}$ Cu, and 0.001–0.450 mg L$^{-1}$ Zn | Non-interactive | [69] |
| Cu + Pb + Zn | *Scenedesmus quadricauda* | 8 | 0.1–0.2 mg L$^{-1}$ Cu, 0.3–0.5 mg L$^{-1}$ Zn, 0.3–0.6 mg L$^{-1}$ Pb | Synergistic (growth) | [81] |
| Cu + Pb + Zn | *Scenedesmus quadricauda* | 8 | 0.1–0.2 mg L$^{-1}$ Cu, 0.3–0.5 mg L$^{-1}$ Zn, 0.3–0.6 mg L$^{-1}$ Pb | Antagonistic (photosynthesis) | [81] |
| Cu + Ti + Zn (nanoparticles) | *Pseudokirchneriella subcapitata* | 7.5–8 | 380 mg L$^{-1}$ TiO, 0.068 mg L$^{-1}$ ZnO, 6.400 mg L$^{-1}$ CuO | Non-interactive | [82] |
| Cd + Co + Fe + Zn + P | *Chlamydomonas reinhardtii* | 7 | 1–100 µM P, 5–40 µM CdCl$_2$ | Antagonistic | [83] |
| Cd + Cu + Ni + Zn | Nile river algal community | NA | 0.05 mg L$^{-1}$ Cd and 0.10 mg L$^{-1}$ Cu, Cr, Zn | Synergistic | [78] |
| Cd + Cu + Ni + Zn | *Pseudokirchneriella subcapitata* | | 0.0200 mg L$^{-1}$ Zn + 0.0010 mg L$^{-1}$ Ni + 0.0025 mg L$^{-1}$ Cu | Non-interactive | [80] |
| Cd + Cu + Ni + Pb + Zn | *Phaeocystis antarctica; Cryothecomonas armigera* | 7.9 | | Synergistic while Zn behaves antagonistic | [84] |
| Cd + Cu + Pb + Zn | *Pseudokirchneriella subcapitata* | NA | 30, 60, 120, 250 and 500 mg L$^{-1}$ for Cd and Zn; and 500, 1000, 2000, 3000, 4000 mg L$^{-1}$ for Cu and Pb | Exude formation lowers metal toxicity | [85] |
| Co + Cu + Fe + Mn + Mo + Ni + Zn | Marine phytoplankton communities | 8.1 | Various oceanic conc. comparison | Complex interactions with biogeochemical influence of ocean | [86] |
| Fe + Cr + Cd | *Micrasterias denticulata* | Near 7 with added soil with buffering property | 600 nM Cd, 10 µM Cr, and 100 µM Fe | Antagonistic | [87] |
| Zn + Cd + Cr | *Micrasterias denticulata* | Near 7 with added soil with buffering property | 600 nM Cd, 10 µM Cr, and 300 nM Zn | Antagonistic | [87] |

Conversion of 1 µM (microM) of (semi)metal for the mentioned elements into mg L$^{-1}$ is following: 1 µM Al = 0.0270 mg L$^{-1}$ Al; 1 µM As = 0.0749 mg L$^{-1}$ As; 1 µM Cd = 0.1124 mg L$^{-1}$ Cd; 1 µM Co = 0.0589 mg L$^{-1}$ Co; 1 µM Cr = 0.0520 mg L$^{-1}$ Cr; 1 µM Cu = 0.0636 mg L$^{-1}$ Cu; 1 µM Fe = 0.0559 mg L$^{-1}$ Fe; 1 µM Mn = 0.0549 mg L$^{-1}$ Mn; 1 µM Ni = 0.0587 mg L$^{-1}$ Ni; 1 µM Pb = 0.2072 mg L$^{-1}$ Pb; 1 µM Se = 0.0790 mg L$^{-1}$ Se; 1 µM Ti = 0.0479 mg L$^{-1}$ Ti; 1 µM Zn = 0.0654 mg L$^{-1}$ Zn.

In a study on Nile phytoplankton community (Chlorophyceae, Bacillariophyceae, and Cyanophyceae) exposed to Co, Zn, and Cu [79], it was observed that lower doses of these metals increased the growth and number of phytoplankton. With a moderate and higher dose of heavy metals, growth inhibition and pigment reduction were induced in the algal

cells. Co showed the lowest toxicity towards the phytoplankton. The research revealed that exposure to the same metallic mixture in the same ratio, dosage, and concentration can result in different stress responses in different phytoplankton, depending on their tolerance level, metabolic activities, and physiology. In a similar study on the Nile River algal community, the effects of Cu, Cd, and Cr were studied using the algal assay procedure [54]. The results revealed that a Cd concentration greater than 1 mg L$^{-1}$ significantly inhibited algal growth. In combined form, all three metals showed synergistic interaction (Table 2). More synergism was noted by Cu + Cd than Cd + Cr. The toxicological effects of the metal pair were in the order of Cd + Cu > Cd + Cr > Cr + Cu.

Several metal toxicological studies have been conducted using the green alga *C. vulgaris*, which provides variable results even with the same pairs, as shown in Table 2. The species is well-studied because it is a model organism of easily culturable species that gives reproducible results in standardised tests recommended by, e.g., OECD, U.S. EPA, EN ISO, or ABNT Brazil norms. Bajguz [88] studied the reciprocal effects of three toxic metals (Cd, Cu, and Pb) on *C. vulgaris*. Under stress after 48 h of cultivation, algal cells showed a decreased chlorophyll content, growth suppression, declined proteins and monosaccharide levels. Cu induced the highest inhibitory effect. In another study, the toxicity of Cd, Co, and Cu cations at EC50 was observed in *C. vulgaris* after 96 h of exposure to the growth inhibition parameter [53]. The cellular response of the green algae was not uniform for different metal combinations, which restates that the mixture toxicity of a metal is altered and it is not what was predicted before. The Cd + Cu and Cu + Co mixtures showed synergistic behaviour. The Cd + Co and Cd + Cu + Co metal combinations acted antagonistically on the given algae. Lam et al. [59], at different pH levels, investigated the effects of Cu and Cd on *C. vulgaris.* With increasing the dose of Cd and Cu, inhibition of algal growth was observed. Growth inhibition with Cu presence was higher at a high pH value. At 1.02 and 4.01 mg L$^{-1}$ of Cd and Cu, the value of EC50 was noted in the green algae. At low doses of Cd + Cu (2.5 mg L$^{-1}$ Cd + <4 mg L$^{-1}$ Cu), a high level of growth inhibition was declared. At high doses of Cd + Cu (2.5 mg L$^{-1}$ Cd + >4 mg L$^{-1}$ Cu), a lower growth inhibition was observed compared to individual Cd and Cu growth inhibition at similar concentrations, which points out a strong antagonistic effect of Cu over Cd that inhibits the stress effect of Cd metal on the test species.

The effect of the toxic metals Cd (0.05 mg L$^{-1}$), Zn, Ni, and Cu (1 mg L$^{-1}$) was investigated by Shehata et al. [78] on Nile water phytoplanktonic community, including blue-green, green algae, and diatoms. The behaviour of these metals in mixtures is indicated in Table 2. The Cd + Ni + Zn metal triplicate reduced algal growth and the amount of chlorophyll *a* decreased from 14.6 mg L$^{-1}$ to 9.64 mg L$^{-1}$ in period of 14 days. After this initial toxicity effect, the algal cells re-established and the chlorophyll *a* content increased to 11.9 mg L$^{-1}$. In the 12-day experimental period, the Cd + Cr + Cu metal mixture showed algal photosynthetic inhibition with 19 mg L$^{-1}$ of chlorophyll *a* at the end of the experiment. The combination of Zn + Ni + Cu + Cd (0.05 mg L$^{-1}$ Cd and 0.1 mg L$^{-1}$ of each Cu, Cr, Zn, and Ni) inhibited algal growth more significantly with a higher decline in the rate of photosynthesis. In this study, a change in the structure of the algal community was also observed after metal application. *Scenedesmus quadricauda*, *Staurastrum paradoxum*, and blue-green alga *Oscillatoria mougeotii* showed the highest resistance against the applied metal combinations.

The reciprocal toxicity of the Cu, Zn, and Ni mixture was investigated in the natural phytoplankton (55 phytoplankton taxa) and zooplankton community of a freshwater ecosystem [89]. For a period of eight weeks, planktonic samples were cultured in a mixture of the three heavy metals (Zn, Ni, and Cu), with variation in doses making a microcosm experiment [89]. At a lower dose of Zn (0.024 mg L$^{-1}$), Cu (0.002 mg L$^{-1}$), and Ni (0.009 mg L$^{-1}$), no prominent effect was observed on plankton, but with higher doses and concentrations (up to 0.404 mg L$^{-1}$ Zn, 0.154 mg L$^{-1}$ Ni, and 0.032 mg L$^{-1}$ Cu), effects on population and community were noticed on the tested species. Multi-substances potentially affected the fraction (msPAF) values for various increasing dosages of Cu, Zn, and Ni, and

six mixture treatments were determined by using the **concentration addition model to bioavailability-normalised single-metal species sensitivity distributions** (SSDs). The three major effects noted were (1) effects at the population level, such as species abundance; (2) effects at the community level, such as species richness, diversity, and composition; and (3) functional effects, such as community respiration rate. Each metallic mixture exposed to the growth, composition, structure, and other factors of the species of the phytoplankton community showed variation. Reciprocal metal toxicity for pairs was also noted in the study, again providing complicated results for each dose and phytoplankton. This indicates that, in mixtures, the toxicity and effects of metals are usually difficult to determine and are much different compared to single metal toxicity on the living cell.

Zhang et al. [65] used *Scenedesmus obliquus* to study Pb, Cd, and Zn ecotoxicology in 96 h experiments. Cd showed higher toxicity than Pb and Zn at the same concentrations. The pairs of Pb + Cd and Zn + Cd showed a synergistic effect, while the Zn + Pb pair resulted in additive toxicity on the alga using an Equivalent–Effect concentration ratio design. This shows that Zn has a different metabolic pathway from Pb and Cd, as also indicated in Table 2. *Scenedesmus quadricauda* showed synergistic toxicity exerted by the Cu, Pb, and Zn mixture on algal growth. The metal mixture exerted an antagonistic effect on photosynthesis [81]. In 42 and 72 h experiments, stress against Cu, Cd, and Zn was noted in *Chlorella* sp. [29] The equitoxic ratio of Cd + Cu acted synergistically, while Cu + Zn, Cd + Zn, and Cu + Cd + Zn acted antagonistically. The highest growth decline was seen against Cd + Cu, indicating that Cd enhanced Cu uptake in the algae. All three metals share the same metal-binding functional groups, such as carboxyl, amino, phosphoryl, hydroxyl, and carbonyl, which have a high affinity for various metal ions, including Cd, Cu, and Zn in algae. In similar research with *Chlorella pyrenoidosa* against Cd (up to 0.6 mg $L^{-1}$) and Cu (up to 1.5 mg $L^{-1}$) [58], it was observed that Cd was accumulated in metal-rich cell granules (up to 98%), and Cu in metal-rich granules (up to 80%) and heat-stable proteins (up to 24%). This indicated that Cu and Cd share the same cellular pathway and binding site for the given microalga. In 72 h exposure against Cu, Ti, and Zn in *Pseudokirchneriella subcapitata* [82], the CuO EC50 value for growth was noted as being the highest, but toxicity on ZnO was highest for algal cells (pigments). The high sensitivity to ZnO and CuO of the algae was attributed to soluble metal ions that are released from the metal oxide particles in the culture medium. Koukal et al. [85], in their work on green alga *P. subcapitata*, noted a high inhibition of photosynthesis when exposed to a lethal dose of Zn, Cd (0.03, 0.06, 0.12, 0.25, 0.5 mg $L^{-1}$), and Cu, Pb (0.5, 1,2, 3, 4 mg $L^{-1}$) for 1 h. A significant decrease in metal toxicity was observed in the alga after the formation of exudates, retaining cell photosynthesis after some time. This indicates that, while making reciprocal metal toxicity risk assessments, it is important to consider possible metal complexation by exudates, lowering metal toxicity [90].

The toxicological effects of Al and Zn were studied in a *Raphidocelis subcapitata* [49]. Zn exhibited 70 times more toxicity on algal cell morphology, photosynthetic pigments, and growth as compared to Al. The results obtained were mainly fitted to the concentration addition model and the dose-level dependence deviation model. Zn and Al behaved synergistically at low doses (below 0.026 mg $L^{-1}$ for Zn and 0.739 mg $L^{-1}$ for Al) and antagonistically at higher. *Desmodesmus quadricauda* exposed to an As and Se binary mixture showed synergistic toxicity [16] (Table 2). The TBARS concentration in the algal cell increased about 70 times in the presence of the As+Se mixture as compared to the control cells, and damage in cell membranes was seen. Se uptake was enhanced by As in the algae. Using the biotic ligand model (BLM), effects of trace elements on the uptake of heavy metal were noted in *C. reinhardtii* [52]. $Ca^{2+}$ exposure decreased Cd uptake. The high ($10^{-5}$ M) and low ($10^{-11}$ M) concentrations of free $Co^{2+}$ did not influence the Cd uptake by the alga. Zn was proven to be the most influential on Cd uptake by the algae. Nagai and Kamo [33] employed a modified biotic ligand model (BLM) to understand the combined toxicity of Zn, Cu, and Cd in the green alga *P. subcapitata*. The Cd + Zn mixture showed an

antagonistic effect on the alga. Zn decreased Cu toxicity in the mixture and the Cd + Cu mixture also showed synergistic interactions, as shown in Table 2.

The reciprocal effects of Cu, Zn, and Ni were investigated in the phytoplanktonic algae *Pseudokirchneriella subcapitata* (*Raphidocelis subcapitata*) [69]. The Cu + Zn + Ni mixture and water samples in all tests with binary mixture metals acted non-interactively on the algal growth. It was concluded that all three metals in combination acted non-interactively on the algae. This indicated that all three metals might have different modes of action on the studied algae. Damage in the algal cells was predominant when a higher dosage of metals was applied to them. Nys et al. [80], in a similar study, determined the combined toxicity of silico–metal mixtures (for Ni, Zn, Cu, and Cd,). Ni + Zn, Cu + Cd, Ni + Zn + Cu, and Ni + Zn + Cu + Cd mixtures were applied to *Pseudokirchneriella subcapitata* and the growth rate was recorded over a period of 48 h and 7 days in *Ceriodaphnia dubia* using a ray design. The Ni + Zn + Cu mixture showed an antagonistic effect on *C. dubia* reproduction. The addition of Cd resulted in non-interactive behaviour of all four metals on *C. dubia.* In the case of *P. subcapitata*, the metal mixture showed non-interactive behaviour, as shown in Table 2. All metal mixtures exerted varying levels of toxicity on both studied species, which predicts that metal toxicity in nature is also species-sensitive. It was observed that the CA (concentration addition) model determined the combined metal toxicity to be 1.2 times higher than normal, overestimating the risk assessment. The study concluded that the IA (independent action) model provided the most accurate results in real time for metal mixtures in fresh-water samples. *Chlorella pyrenoidosa* 251 exposed to a Cd, Cu, and Ni mixture showed synergistic effects [91]. A high decline in growth, pigments, and photosynthesis was noted.

Koppel et al. [73] studied the reciprocal toxicity of Cd, Co, Ni, Pb, and Zn in *Phaeocystis antarctica* and *Cryothecomonas armigera.* The results showed that Cu in the metal mixture was the primary factor of growth inhibition of a rate with $R^2$ values > −0.84 for all cellular fractions in both algae. The bioaccumulation of all metals increased with increasing concentrations, except for Ni and Zn. Zn exhibited an antagonistic interaction with Cu in this investigation, shown in Table 2. $Cd^{2+}$ was added to *Thalassiosira weissflogii* culture medium at a high concentration [63] and, as expected, growth inhibition and cell damage were induced. The addition of EDTA and iron ions caused a notable decline in Cd toxicity. The high ferric ion activity was concluded to be the reason for the neutralisation of Cd toxicity. Antagonistic behaviour between Cd and Fe was observed, as they share a binding site on algal cell membranes at a concentration of $10^{-6.8}$ M Fe and $10^{-5}$ M of EDTA in the culture. Volland et al. [87] also explained the protective effect of divalent ions of Fe, Zn, and Ca on the toxicity of Cd, Cr, and Pb on *Micrasterias*. Analytical transmission electron microscopic (TEM), electron energy loss spectroscopy (EELS), and electron spectroscopic imaging (ESI) showed that Cr metal uptake by the algal cell was decreased when the alga was pre-treated with Fe ions (5.584 mg L$^{-1}$). Zn decreased Cr toxicity by promoting cell respiration, the rate of photosynthesis, and growth. Fe and Zn and Ca and Fe lowered Cr stress by occupying the same cellular binding sites in competition with Cr. In a similar experiment, it was observed that pre-treated cells of *Micrasterias* with $CaSO_4$, when exposed to Cd (up to 16 mg L$^{-1}$), decreased Cd stress effects in terms of photosynthesis rate. Cell ultrastructural damage was also controlled [70]. Marine diatom *Asterionella japonica* Cleve was tested against Cu, Cd, Pb, H, Zn, and Mn [51]. The relative toxicity of all metals showed a high correlation (r = 0.961) in the form of metal sulphides. Toxicity might be induced in cells by these metals (single and combined) through binding to sulfhydryl-containing compounds in the cell membrane of diatoms.

In an experiment on *Chaetoceros gracilis* and *Isochrysis* sp. marine microalga, the reciprocal toxicity of Cd and Cu was determined [55]. Using the IC50 test for 96 h to estimate growth inhibition, it was reported that Cd (IC50 = 2.370 mg L$^{-1}$) was more toxic to *C. gracilis* than to the growth of *Isochrysis* sp. (IC50 = 490 mg L$^{-1}$). For Cu concentrations of 63.75 mg L$^{-1}$, in *C. gracilis* and for *Isochrysis* sp., the Cu toxicity was noted at 31.80 mg L$^{-1}$. Both Cd and Cu, in binary exposure, synergistically caused growth inhibition

in both marine algae (Table 2). Braek et al. [72] observed synergistic toxicity of Cu+Zn in all test strains (3 marine diatoms and 1 dinoflagellate), except *Phaeodactylum tricornutum* Bohlin. It was assumed that Zn ions in the mixture decreased the growth of *P. tricornutum*. Mg application (low concentration) enhanced Zn stress in the algae, maybe because of a common mode of action. In a similar study, Zn and Cd in a mixture exerted a synergistic effect on the growth of *P. tricornutum* and *Skeletonema costatum* [67]. Cd was found to be more toxic to *S. costatum* and Zn showed a higher toxicity to *P. tricornutum* due to its different tolerance capacities and metabolic dissimilarities. The reciprocal effect of marine microalgal species *Phaeocystis antarctica* and *Cryothecomonas armigera* against Cd, Cu, Ni, Pb, and Zn was studied [84]. Both algae showed an acute toxicological response upon exposure, which could be detected as growth inhibition (up to 10–54% by the equitoxic metal mixture and 5–10% by the environmental metal ratio mixture). A decline in chlorophyll *a*, cell size increase, and lipid accumulation were seen against metal mixtures. Both species showed similar stress responses, as stated in Table 2, to the equitoxic mixture, noninteractive by IA and antagonistic by the CA model. The prepared metal mixture revealed antagonistic interactions by IA with both alga, and antagonistic behaviour of the metals was observed at low doses and synergistic behaviour at high doses by both the IA and CA models. This study used the IA and CA models along with EC10 values to understand the combined metal toxicity on algae, which varied species-wise. An antagonistic effect between Cu and Cd was reported in *Dunaliella minuta* [62], a marine Chlorophyta reported in Table 2. The alga was exposed to 0.492 mg L$^{-1}$ Cu and 0.038 mg L$^{-1}$ Cd in a binary mixture for 96 h, which resulted in 50% growth inhibition and pigment decline in the alga.

The antagonistic toxicological effects of Cu + Cd were observed in *C. reinhardtii* [56]. High Cu exposure reduced glutathione levels in the cell by inhibiting GR enzyme activity. In contrast, Cd enhanced the glutathione levels in the cell by increasing GR enzyme activity. Measuring glutathione changes after metal exposure is a good, non-invasive technique used to study the toxicity of any metal. Xie et al. [31] also proved the antagonistic behaviour of the Cd and Cu mixture using the DEBtox model (Dynamic Energy Budget toxicology model) on *C. reinhardtii*. The DEBtox model overestimated the combined toxicity of Cu + Cd because of the antagonistic effect. In another study with *C. reinhardtii* [71], it was noted that the uptake of Pb was reduced by 87% by adding 0.031 mg L$^{-1}$ of Cu to the mixture, about 50 times higher in concentration compared to Pb addition up to 0.002 mg L$^{-1}$. This might suggest common pathway or receptors for Pb and Cu. In a similar study, it was reported that the uptake of Cd was highly influenced by the concentration of Zn in the media on *C. reinhardtii* [66]. A concentration of $2 \times 10^{-8}$ M of Zn$^{2+}$ added to the algal culture regulated the uptake of increasing doses of Cd$^{2+}$ (up to $1 \times 10^{-8}$ M). A strong inhibitory effect was observed in Fe-containing media, which lowered Cu toxicity by up to 50% (0.00012 mg L$^{-1}$ to 0.00006 mg L$^{-1}$) after 72 h of exposure in *C. reinhardtii* [68]. This indicates that Fe and Cu might share a similar mode of action or cell binding site. In another investigation, trace metal nutrients essential for growth, such as Fe, Zn, Co, Mn, Cu, Ni, and Mo, were investigated for their phytoplankton interaction in ocean water [86], proving that trace metals can alter the mixture toxicity of elements, including heavy metals. Wang and Dei [57] observed that the rate of accumulation of Cd and Cu increased in *C. reinhardtii* with increasing concentrations of phosphorous and free metal ions for Cu and Cd. Most Cd accumulation was in the chloroplast, inhibiting photosynthesis. Cu affected the electron transport chain and photosystems. It was observed that P-deficient conditions in the medium caused more cellular damage to green algae when exposed to Cu and Cd acting synergistically (Table 2).

Ou-Yang et al. [74] examined the impact of sub-lethal doses of Cu, Cr, Zn, Cd, and Pb on *C. vulgaris* in a 96 h exposure test. All metals up to 5 mg L$^{-1}$ showed high growth inhibition. The toxicity effect weakened as the duration of exposure increased. Cu and Cr induced an inhibiting effect, and Zn and Cd showed a promoting effect on chlorophyll content. On growth and photosynthesis, all the metals exerted effects independently of each other. Stress effects of Cd and Cr were observed in *C. vulgaris* [60]. Both metals

caused an increase ROS and growth, and decline in pigments. The real-time PCR results obtained reported that Cu and Cd acted independently to inhibit PSII activity and $CO_2$ integration in algae. Cu (0.095 mg $L^{-1}$) + Cd (0.224 mg $L^{-1}$) increased the ROS content in the cell by approximately 9 times as compared to a control. A real-time PCR evaluation revealed that Cu + Cd reduced the abundance of photosystem II protein D1 and ribulose bisphosphate carboxylase/oxygenase genes transcripts in the green alga. Cr toxicity was observed against different concentrations of applied phosphorus in *C. vulgaris* [92]. Low P (10 μM) in the culture media resulted in a high inhibition of growth in algal cells by Cr (48 and 96 h of exposure). P is an essential element for growth and is always present in cultures, which might hinder/alleviate the effect of Cr and other metals, as reported in Table 2. Webster et al. [83], in their 7-day experiment with *C. reinhardtii*, explained that, in low P (0.01 mM) conditions, algae are more prone to be affected by Cd. Under P (0.1–1 M)-rich conditions, algal cells accumulated essential trace elements (Co, Fe, K, Zn, and Na), which interfered with Cd (up to 15 μM) binding, resulting in low Cd toxicity toward *C. reinhardtii*. Rodgher et al. [75] confirmed the role of Pin Zn toxicity on *R. subcapitata*. A low availability of P (6.5 mg $L^{-1}$) acted additively with Zn and caused more damage to the alga compared to individual stress effects. A moderate to high concentration of P (up to 30 mg $L^{-1}$) helped reduce Zn toxicity to the algae. Microalgae require a variety of nutrients for optimal growth and health, and P is one of the most essential limiting factors for algal growth. As explained above, many studies have confirmed that, under P-limited conditions, metal stress effects are higher in algal species [93]. P is a part of algal nucleic acids, adenosine triphosphate, and phospholipid membranes. P plays an important role in enzymatic synthesis and energy transfer in photosynthesis. The available P is directly proportional to the growth rate. The importance of P for microalgae is confirmed by their ability to store P. This stored P is released under P-limiting conditions by phosphatase enzymes. As a result, organic bound phosphates are released, which can be used by algae for homeostasis [93].

## 6. Possible Mechanisms of Entry, Toxicity, and Detoxification in Algal Cells

For the reciprocal effects seen in marine and freshwater algal populations, we reported in Table 3 the articles in which antagonism, synergism, and additive reciprocal effects were observed. Unlike marine algae (Table 3), a combination of metal ions was also observed between freshwater species without a confirmed interaction between them. Only additive and synergistic effects were confirmed in marine algae for Hg (in the case with Ni or Cd), while for the combination of Cd + Cu, no additive effects were calculated as far as we know. An interesting study was the test of the Cd + Cr + Ni + Pb mixture in the presence of Zn [84], where these toxic elements had a reciprocal synergistic effect on Antarctic marine microalgae, while Zn acted as an antagonist of this mixture. It seems that, for both kinds of algae (freshwater and marine), there are more articles with presumed antagonism than a synergism and/or additive effect.

**Table 3.** Reciprocal effects of metal mixtures in marine and freshwater algal populations.

| Antagonism | Synergism | Additive Effect | Non-Interactive |
|---|---|---|---|
| Marine algal populations: | | | |
| Cd + Cu [32] | Cd + Cu [55] | Cd + Zn [67] | |
| Cd + Fe [63] | Cd + Hg [64] | Cu + Zn [32] | |
| Cd + Ni [64] | Cd + Ni [64] | Hg + Ni [64] | |
| Cd + Cr + Fe [87] | Cd + Cr + Ni + Pb + Zn [84]—except of Zn in the mixture | | |
| Cd + Cr + Zn [87] | Cu + Zn [72] | | |

**Table 3.** *Cont.*

| Antagonism | Synergism | Additive Effect | Non-Interactive |
|---|---|---|---|
| Cd + Cr + Ni + Pb + Zn [84]—Zn as antagonist to others | | | |
| Cd + Cu + Ni + Pb + As(V) [76]—As(V) has the main toxicity | | | |
| Cu + Zn [73] | | | |
| Cu + Co + Fe + Mn + Mo + Ni + Zn [86]—complex interactions with the biogeochemical part of ocean | | | |
| Freshwater algal populations: | | | |
| Al + Zn [49,50] | As + Se [16] | Cu + Cr [74] | Cd + Co [52] |
| Cd + Ca [70] | Cd + Co [77]—low Cd and high Co | Zn + P [75] | Cd + Cu + Fe + Mn [52] |
| Cd + Co [53] | Cd + Cr [54] | Zn + Pb [65] | Cd + Cu + Ni + Zn [80] |
| Cd + Co [77]—high Cd and low Co | Cd + Cu [29,33,53,54,57,58,60,61] | | Cu + Ni [69] |
| Cd + Cu [31,56,59] | Cd + Pb [65] | | Cu + Ni + Zn [80] |
| Cd + Zn [24,29,52,66] | Cd + Zn [65] | | Cu + Ti + Zn (nanoparticles) [80] |
| Cd + Co + Cu [53] | Cd + Cu + Ni + Zn [78] | | |
| Cd + Cr + Cu [78] | Cu + Co [53] | | |
| Cd + Ni + Zn [78] | Cu + Zn [19] | | |
| Cd + Co + Fe + Zn + P [83] | Cu + Cr + Ni [51] | | |
| Cu + Fe [68] | Cu + Pb + Zn [81]—for growth | | |
| Cu + Pb [71] | | | |
| Cu + Zn [29,33] | | | |
| Cu + Pb + Zn [81]—for photosynthesis | | | |

However, there are other relevant factors that can affect the toxicity of metals, such as the different sensitivities of algal species, saline vs. freshwater with lower concentrations of salts, which can influence chemical reactions and others mentioned earlier. One of them is seasonal variations, which are being studied by more scientists. Temperature can significantly change the behaviour of algae also during seasons throughout the whole year. Seasonal variations in algae were confirmed for Ni and Fe in *Halimeda tuna* and for Fe in *Pteroclaudia pinnata* [94]. Moreover, Cd and Pb were generally lowest in spring algae samples [94]. Seasonal variations in metals in brown seaweed have been observed by several authors, including [95]. These authors observed variations in Cd or Cu in samples collected at different times of the year, with the maximum in February and the minimum in August, while Pb did not copy that seasonal pattern. The maximum of Zn was observed in March and the minimum in September in brown seaweed (*Fucus vesiculosus*). Villares et al. [96] studied two genera of macroalgae, *Ulva* and *Enteromorpha,* and they concluded that the seasonal variations in Al, Cr, Cu, Fe, Mn, Ni, and Zn followed a similar pattern in both seaweeds and appeared to be caused by dilution during the period of maximum growth and concentration during periods of slow growth. Fluvial inputs of Al, Fe, and Mn in autumn and winter also appeared to accentuate the latter effect. The seasonal variations in the biomass of macroalgae were also studied due their related energy content and biomethane potential [97].

The possible mechanism of metal toxicity and the defence system of algal cells is not as studied as in higher plants. It assumes that the entry and detoxification system between plants and algae can be very closely related, as previously described, for the toxicity of

Cd, together with the synthesis of phytochelatins and their ability to absorb heavy metal ions in the cytoplasm of algae cells [98–100]. The initial stage involves passive extracellular adsorption, called biosorption, on the cell surface, and the next stage includes active intracellular diffusion and accumulation, called bioaccumulation. The first component of the microalgal cell that interacts with and binds to heavy metal pollutants is the cell wall due to the abundance of amino, carboxy, sulphate, and hydroxyl groups on its surface [101]. These negatively charged groups allow for the binding of the metal ions on the microalgae cell surface from its surrounding environment. Biosorption achieved due to the presence of various biomolecules on the microalgal cell wall, such as proteins, carbohydrates, and lipids, is aided by negatively charged ions such as phosphate ($-PO_4$), carboxyl ($-COOH$), hydroxyl ($-OH$), and amino ($-NH_2$) groups [101]. In addition to compartmentalisation or efflux, there is another pivotal step in detoxification, namely, the binding of the metal ions with certain peptides and the consequent chelation process [102]. The chief proteins that contribute to this step are metallothioneins and phytochelatins, and they thereby maintain a steady concentration of the up-taken metal ions inside the cell. Transporters also moderate intracellular concentrations accordingly by strategically releasing heavy metals out of the cell [102]. In the case of different metal ions outside of the algal cells, there are interactions at the entry into the cell, where the pH value and chemical properties of the metal ions (e.g., valency, steric properties, and similarities with essential metal ions that are known mostly for bivalent cations) can play a key role in the potential toxicity of metals to cells.

The cation diffusion facilitator (CDFs) proteins form a family of ubiquitous transporters involved in metal homeostasis and tolerance. These proteins catalyse the efflux of transition metal cations, such as $Zn^{2+}$, $Cd^{2+}$, $Co^{2+}$, $Ni^{2+}$, or $Mn^{2+}$, from the cytoplasm to the outside of the cell or into subcellular compartments [103]. Hanikenne et al. [103] identified metal tolerance proteins (MTPs) in the genome sequences of *C. reinhardtii* and *Cyanidioschizon merolae*—CrMTP1 and CmMTP1 (both in vacuole)—that are related to the Zn transporters of higher plants, humans, and, more distantly, yeasts, which may play the same role in the algal cells [104]. They found algal Irt-like proteins (ZIPs), mainly in the vacuolar and plasma membrane, which regulate Zn and Fe uptake. Together with these findings, they identified the vacuolar single Cu transporter (COPT) protein in the studied algal genomes that regulates Cu uptake by living cells; the heavy-metal P-type AT-Pases (HMAs) in organelles with the regulation of either monovalent $Cu^+/Ag^+$ or bivalent $Zn^{2+}/Co^{2+}/Cd^{2+}/Pb^{2+}$ cations; and vacuolar/organell MRPs helping to regulate Cd, As, and Hg in the cell. Moreover, they observed the presence of natural resistance-associated macrophage proteins (NRAMP) in vacuoles that transport a wide range of bivalent cations and have been more specifically implicated in Mn, Cu, and, more marginally, Fe homeostasis [103]. Other genes for organellar ATM/HMTs in genomes of *C. reinhardtii* and *C. merolae* regulating the cell Fe and S concentrations are half-size transporters located either in the mitochondrial or vacuolar membrane. CAX proteins are divalent cation/$H^+$ antiporters in the transmembrane area that are expressed in stress (CrCAX1).

Other authors [104] identified genes encoding multicopper oxidase (CrFOX1) and Fe permease (CrFTR1) in *C. reinhardtii* that occur in the Fe assimilation pathway. The Fe status in *Chlamydomonas* is regulated by genes of the ZIP family, such as *IRT1* and *IRT2* [105]. These algae species also encode members of the NRAMP family, which are also involved in Fe regulation [106]. Animal-like transferrin was first discovered in the unicellular green alga *Dunaliella salina* as a salt-induced 150 kDa plasma membrane bound protein and was initially called p150 for Fe regulation [107]. The CDF protein family MTPs in *Chlamydomonas* are predicted to transport [108]. Although *Aureoreococcus anophagefferens* does not have a plastocyanin gene, the abundance of Cu-dependent proteins in the proteome is unknown [109]. All other algal genomes have at least one CTR gene to regulate Cu [109].

The tolerance of *Chlorella* sp. to Cr and Pb is probably caused by the reduction in metal influx across the plasma membrane; metal chelation in the cytosol by ligands (e.g., phytochelatins, metallothionein, organic acids, and amino acids); the transport of metal ligand complexes through the tonoplast and accumulation in the vacuole; sequestra-

tion in the vacuole by tonoplast transporters; and/or ROS defence mechanisms [110]. S is essential for the synthesis of important defence compounds, including metalothioneins, oxidised glutathione, and phytochelatins. However, Cr(VI) can induce a sort of S-starvation by competing for uptake with sulphate and causing the depletion of reduced cellular compounds. Ferrari et al. [111] studied $H^+/(SO_4)^{2-}$ (*SULTRs*) and $Na^+/(SO_4)^{2-}$ (*SULPs*) plasma membrane sulfate transporters and a chloroplast-envelope localised ABC-type holo-complex in *Scenedesmus acutus* with different Cr(VI) sensitivities, and they observed that the *SULTRs'* up-regulation, observed after S-starvation, may directly contribute to enhancing Cr tolerance by limiting Cr(VI) uptake and increasing S availability for the synthesis of S-containing defence molecules. Adenosine-triphosphate (ATP) binding cassette transporters (ABC transporters), one of the largest and oldest gene families, are responsible for the translocation of many substrates across membranes [112]. They are the most abundant metal transporter families in all phytoplankton, which typically transport more than one type of metal, such as the Ni/Co uptake transporter family and the Mn/Zn/Fe chelate uptake transporter family [112].

Furthermore, phytohormones, such as auxins and cytokinins, play a role in the integration of growth control and stress response, but their role in adaptation to heavy metal remains to be elucidated [113]. Current research has indicated that Pb, one of the most toxic metals in nature, causes severe depletion of endogenous cytokinins, auxins, and gibberellin and an increase in abscisic acid content in the green alga *Acutodesmus obliquus*. Exogenous auxins and cytokinins alleviate Pb toxicity through the regulation of endogenous phytohormone levels [113]. The presence of Hg/Cu/Zn in a sensitive strain of *Chlorella sorokiniana* represents competition in the synthesis of metallothionein, even detecting a greater increase in them [114]. Cd and Co induced growth and photosynthetic inhibition in *Raphidocelis subcapitata* [77]. Antagonism occurs with Cd and Co because they probably compete for the same transport sites on the membrane, since they are bivalent metals [77].

The tolerance of microalgae to metal toxicity with a high supply of P is probably because microorganisms supplied with this nutrient resist metal toxicity better compared to algal cells under severely limited P conditions [75]. In a P-rich environment, microalgae have been reported to incorporate phosphorous as polyphosphate granules, and these granules can bind metals, protecting microorganisms from metal toxicity [75]. The divalent ions of Fe, Zn, and Ca were able to diminish the effects of Cd, Cr, and Pb on *Micrasterias* [87]. Cd is taken up by Ca and Fe transporters, whereas Cr appears to enter algae cells via Fe and Zn carriers. It was shown that Pb is not taken up in *Micrasterias* at all, but exerts its adverse effects on cell growth by substituting Ca for the cell-wall-bound Ca.

## 7. Conclusions

A vast number of waste products, including toxic metals, are released into the aquatic environment, leading to chemical mixtures in aquatic systems and reciprocal effects among them. However, current toxicological assessments mainly focus on single metal toxicity. To fully understand the potential threat of metals, understanding mixture toxicity in an aquatic ecosystem is vital. The use of phytoplankton can be very useful in ecotoxicological studies, as they make up the first level of any aquatic food web, with the primary responsibility for the transformation of metals in aquatic systems and interactons with biogeochemical cycles in lakes and oceans. The recent methodologies used to address this issue have some shortcomings. The EC50 value, the community response, microtox assays, BLM, CA, IA, and other statistical methods provide valuable predictions, but have their shortcomings. In the review, it was noted that most metal toxicity research is conducted with Al, Cd, Co, Cr, Cu, Fe, Ni, Pb, and Zn metals and their reciprocal effects on increased or decreased toxicity. Cu mostly showed antagonistic effects with Cd, Pb, and Zn. The Zn metal ion reduced the toxicity of Al, Cd, Cu, and Ni in many studies. The Cd metal showed variable interactions in different species with Cu and other metals. This suggests that, when studying these metals on phytoplankton, it is important to consider the possible reciprocal interactions of these metals in a mixture. While Zn in the mixture of Cd + Cr + Ni + Pb

reacts in the marine algal population as an antagonist, all the mentioned metals can be assessed as synergistic reciprocal effects. We did not find articles with marine algae that describe the non-interactive effects among metals, while, for freshwater algae, there are more of them. This may be because freshwater algae are more often studied compared to marine algae. Antagonism in freshwater algae was observed for the combination of Cd + Co, when Cd was high and Co was low. In the opposite situation, synergism was observed. In the next part, we described some relevant parameters (e.g., temperature and seasonal variations in toxicity, interactions with P, and detoxifying systems) that can influence metal toxicity. We mentioned the transporters and channels that can regulate metal homeostasis in organelles and cells such as CDF, ZIP, HMA, COPT, MTP, MRP, and NRAMP that regulate and transport essential and non-essential metals through the membranes of algal cells and organelles. We were more focused on the transporters on the cellular and vacuolar membrane. These proteins are sometimes the target of metal toxicity because of the interactions between essential and non-essential metal ions in front of them and inside the cytoplasm. Some defence mechanisms were described, with a focus on the most useful metalothioneins and phytochelatins. A brief introduction to metal cellular pathways and detoxification in the review can help researchers to understand this metal toxicity in phytoplankton for various metal ions, helping future researchers in the field to understand these possible reciprocal toxic metal interactions that affect metal toxicological assessments, and in planning their experiments. The overview of marine and freshwater phytoplankton metal stress response differences can also be helpful in understanding the underlying differences between the two communities' structure, metabolism, environmental differences, and behaviour. Modernisation cannot be reversed, and it will introduce more complex mixture pollutants with metals. It is important to evaluate the potential risks and work at early stages with modern and more accurate methods, together with computational prediction, to avoid serious damaging situations which will be extremely difficult to evaluate and cure.

**Author Contributions:** A.N. was the main author of the original draft. M.M. supervised and was involved in conceptualization. P.E.Š. partially helped in writing of the original draft. A.N., P.E.Š. and M.M. reviewed and edited the draft versions. All authors have read and agreed to the published version of the manuscript.

**Funding:** This research received no external funding.

**Conflicts of Interest:** The authors declare no conflict of interest.

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
