# Peer review of "Reciprocal Effects of Metal Mixtures on Phytoplankton"

_phycology, doi:10.3390/phycology4010007_

Round 1

Reviewer 1 Report

Comments and Suggestions for Authors

Dear Authors, 

The review manuscript highlighs the reciprocal effects of metals in phytoplankton species. It is a study with high (eco)toxicological relevance, since it brings a synthesis of the knowledge that moves beyond the classical ecotoxicological approach of single chemical effect, to a more realistic approach considering metals mixtures and combinations and their effects on a community commonly used as proxy of disturbs in aquatic ecosystems. Despite the study is relevant for Phycology readers, some writing issues address some inconsistency which must be carefully revised. 

Please, find in attached the manuscript file with the detailed revision, and below some critical points to improve the MS as a whole:

- It is essential to describe the main organisms' groups that comprise the phytoplankton community and their features that would contribute on the sensitivity and/or tolerance to metal stress;

- Many paragraphs are sequential descriptions of resullts from other papers, but a general interpretation with a slight discussion is lacking. The excess of results with very often unnecessary details and lengthy sentences, make the reading exhaustive and less informative;

- Further information should be provided, regaring the cellular and molecular mechanisms of metals action (e.g., cell transporters blocked); metalloids (e.g., Arsenic) effects and, at least, a paragraph highlighting the relevance of metallothioneins on metals detoxification.

- A graphical abstract or flowchart would be very welcome, as usually suitable for review papers, to display overall interactive effects of metals in phytoplankton (both Eukaryotic and Prokaryotic) cells. 

Comments on the Quality of English Language

Dear Authors, 

I recommend the manuscript to undergone language revision. Minor gramatical errors are found along the text, besides some words that could be replaced for more suitable ones.  

Author Response

Dear Reviewer 1, please, see the attachment.

Reviewer 2 Report

Comments and Suggestions for Authors

Review for the paper "Reciprocal Effects of Metal Mixtures on Phytoplankton" by Ammara Nawaz, Pavlína Eliška Šotek and Marianna Molnárová submitted to "Phycology".

General comment.

A growing body of evidence indicates robust physical-biological linkages in marine plankton ecosystems, as exemplified by observed variability across hydrographic interfaces. While metal influences on phytoplankton have been well studied, there has been less investigation into the combined effects of metal mixtures. Recent studies have provided further insight into metal-microalgae interplays. The authors reviewed data summarizing phytoplankton responses to exposures from metal mixtures. While the data presented is of interest, the manuscript would benefit from improved presentation focusing on key results. Additionally, the overall length should be reduced by removing non-essential technical details. The concluding remarks should be expanded to address potential ecological implications arising from the main findings.

Major concerns.

Introduction

- The scientific motivation and novelty of the study should be highlighted in greater detail. Clearly state the specific aim and knowledge gaps being addressed.

Behavior of Heavy Metals in an Aquatic Ecosystem

- Include data on seasonal variations in phytoplankton responses to environmental factors. Specifically emphasize the role of temperature.

- Compare responses of marine versus freshwater microalgae species to environmental variables. Indicate any differential responses observed between the two groups.

Previous Investigations done to Determine Reciprocal Toxicity of Metal Mixtures in Phytoplankton

- Split this lengthy section into subsections grouping findings of similar toxic effects. 

- Shorten the section by removing excessive wordiness and technical details. Include only the most relevant studies.

- Provide interpretation of the previous studies, with an emphasis on potential ecological impacts and implications of the findings.

Conclusion

- Clearly state the key original findings and main takeaways from the current study. Indicate how the results address the aims initially set forth.

Specific remarks.

The authors use the terms cadmium, copper, nickel etc together with Cd, Cu, Ni etc. in the text. I suggest using short acronyms Cd, Ni, Zn in the entire text.

L15. Consider replacing "recherche" with "review".

L72. Consider replacing " Phytoplankton’s " with " Phytoplankton".

L100. Insert full stop after "algae".

L105. Delete "specie".

L119. Consider replacing " there" with "available".

L132, 186, 263, 287, 295, 327, 415, 488, 521. Consider replacing " specie " with " species ".

L148. Consider replacing " Now a day’s scientists are commonly applying statistical approach " with " Now, scientists commonly apply statistical approach ".

L150. Consider replacing " Because for algae is known that toxicity of metals can be " with " It is known that metals toxicity for algae can be ".

L151-152. Consider replacing " coenobia, more authors determine " with " coenobia. Therefore, many authors determine ".

L186. Consider replacing " making " with " makes ".

L197. Consider replacing " Phytoplanktonic " with " Phytoplankton".

L199. Consider replacing " form " with " from the".

Table 2, first row. Consider replacing " specie(s) " with " species ".

Table 2 presents data on metal concentrations expressed in different units (M, nM, mg L-1 etc). It would be useful to provide equations for different units for each metal (e.g. 1 nM Cd = ... mg L-1 Cd). Possibly, it can be made in a supplement file or as a footnote for Table 2.

L225. Consider replacing " Given below in the Table 2 is a summary of these ecotoxicological reciprocal metal assessments " with " Table 2 summarizes ecotoxicological reciprocal metal assessments ".

L231. Consider replacing " water phytoplankton " with " phytoplankton ".

L235. Consider replacing " least " with " lowest ".

L269. Consider replacing "chlorophyl a " with " chlorophyll a", "a " must be in Italic.

L295. " Scenedesmus " must be in Italic.

L304. Correct the reference.

L335. Consider replacing " exposure time " with " exposure ".

L348, 378. Consider replacing " 7 days " with " 7-day ".

L359. Consider replacing " specimen " with " species ".

L411. Consider replacing " The hallmark of the study was using " with " This study used ".

L431. Consider replacing " noted " with " noted as".

L554. Consider replacing " prime reason" with " primary factor".

Comments on the Quality of English Language

Some revisions are required.

Author Response

Dear Reviewer 2, please, see the attachment. 

Round 2

Reviewer 1 Report

Comments and Suggestions for Authors

Dear Authors, 

The manuscript has improved significantly and all of the issues have been correctly addressed. I just want to highlight that the graphical abstract only refer to an eukaryotic cell, although the phytoplankton also comprise prokaryotic organisms (Cyanobacteria). It would be welcome to provide a scheme showing the effects of metals on those cells.